# Branched-Chain Amino Acids and Insulin Resistance, from Protein Supply to Diet-Induced Obesity

**DOI:** 10.3390/nu15010068

**Published:** 2022-12-23

**Authors:** Jean-Pascal De Bandt, Xavier Coumoul, Robert Barouki

**Affiliations:** INSERM UMR-S 1124-T3S, Environmental Toxicity, Therapeutic Targets, Cellular Signaling & Biomarkers, Faculté des Sciences, Université Paris Cité, UFR des Sciences Fondamentales et Biomédicales, 75006 Paris, France

**Keywords:** leucine, valine, isoleucine, type 2 diabetes, glucose tolerance, protein consumption, branched-chain keto acids, acylcarnitine, hydroxyisobutyrate

## Abstract

For more than a decade, there has been a wide debate about the branched-chain amino acids (BCAA) leucine, valine, and isoleucine, with, on the one hand, the supporters of their anabolic effects and, on the other hand, those who suspect them of promoting insulin resistance. Indeed, the role of leucine in the postprandial activation of protein synthesis has been clearly established, even though supplementation studies aimed at taking advantage of this property are rather disappointing. Furthermore, there is ample evidence of an association between the elevation of their plasma concentrations and insulin resistance or the risk of developing type 2 diabetes, although there are many confounding factors, starting with the level of animal protein consumption. After a summary of their metabolism and anabolic properties, we analyze in this review the factors likely to increase the plasma concentrations of BCAAs, including insulin-resistance. After an analysis of supplementation or restriction studies in search of a direct role of BCAAs in insulin resistance, we discuss an indirect role through some of their metabolites: branched-chain keto acids, C3 and C5 acylcarnitines, and hydroxyisobutyrate. Overall, given the importance of insulin in the metabolism of these amino acids, it is very likely that small alterations in insulin sensitivity are responsible for a reduction in their catabolism long before the onset of impaired glucose tolerance.

## 1. Introduction

Branched chain amino acids (BCAAs: leucine, valine, isoleucine) have been under close scrutiny since the beginning of the 1970s because of their unique role in endogenous metabolism and their critical regulatory properties on protein metabolism. A notable property of this metabolism is its compartmentalization; in humans, at mealtime, while more than 50% of ingested amino acids are taken up by the liver, BCAAs essentially escape from the splanchnic territory and directly contribute to the postprandial activation of peripheral/muscle protein synthesis. It was initially thought that the primary role of these amino acids was restricted to the anabolic or anticatabolic properties of leucine. Indeed, this rapidly led, in the clinical setting, to the development of BCAA-enriched nutritional formulas [1] for the dietary management of cirrhotic patients (and hypothetically to treat hepatic encephalopathy, a neurological disorder due to a severe chronic liver disease) [1] or 2) for the stimulation of anabolism or slowdown of catabolism in injured patients [2]. Already at that time, there were some hints of possible interactions between BCAAs and insulin signaling. However, it was only around the years 2005–2010 that several studies suggested, on the one hand, a possible association between type 2 diabetes (T2D) and increased plasma BCAA levels and, on the other hand, a possible causal link between BCAAs and insulin resistance. This review is the result of a long-term scientific vigil, since 2016, specifically on the following keywords: branched-chain amino acid OR leucine OR valine OR isoleucine AND insulin resistance. Studies in which the existence of insulin resistance was not clearly defined on objective biological criteria were excluded. Additional searches on the metabolic properties of BCAA derivatives were carried out a posteriori on the following keywords: insulin resistance AND ketoisocaproate, ketoisovalerate, ketomethylvalerate, hydroxyisobutyrate, acyl-carnitine.

In the present article, we will first review the metabolism and properties of BCAAs. We will then analyze the factors involved in the plasma variations of these amino acids and in particular their interest as markers of insulin resistance. We will then focus on the interrelationship between insulin resistance and BCAAs, first focusing on its consequences on BCAA metabolism before analyzing the novel properties of BCAAs and their interactions with insulin effects. We will attempt to elucidate the complex physiological and species-dependent mechanisms that govern these interactions.

## 2. BCAA Metabolism

### 2.1. BCAA Inter-Organ Exchanges in the Postprandial Period

From the physiologist’s point of view, and leaving aside the digestion and digestive absorption stages, the metabolism of amino acids and more particularly of essential amino acids such as BCAAs, starts with inter-organ exchanges in the postprandial period. Ten Have et al. [3] measured net BCAA and metabolite fluxes across liver, portal drained viscera, kidney, and hindquarter in a multi-catheterized pig model receiving a test meal (30% of the daily intake as a bolus). In the postprandial phase, BCAAs were significantly metabolized by the portal-drained viscera (on average, 57% of leucine, 50% of isoleucine, and 60% of valine), producing through the action of transaminases branched-chain keto acids (BCKAs), which were then used by the liver. They further observed that BCAAs escaping splanchnic metabolism then underwent an uptake, essentially by the hindquarter and the kidneys. One thing to keep in mind about this muscle peripheral utilization is that it depends in part on the insulin-dependent increase in blood flow that will condition the peripheral availability of substrates. In addition, the relative flux of BCAAs in liver vs. muscle is species dependent. It is important to note that BCAA metabolism significantly differs quantitatively between rodents and humans: in humans, transaminative and oxidative capacities are higher (66% of total capacities) in the skeletal muscles. In rats, oxidation is mainly hepatic (60%) and total body oxidative capacity is nearly 10-fold higher compared to humans [4].

The interorgan compartmentation of BCAA metabolism is such that, after a protein meal, BCAAs represent a major part of the amino acids released by the splanchnic bed. Indeed, Wahren et al. [5] showed in healthy volunteers that, after an oral protein load (3 g/kg), BCAAs accounted for more than half of the amino acids released by the splanchnic area at 60 min after the protein meal and were the only amino acids consistently released at 120–180 min. This was associated with a respective 2.5, 1.9, and 3.3 increase in plasma leucine, valine, and isoleucine levels.

### 2.2. BCAA Celllular Metabolism

#### 2.2.1. BCAA Transport

At the cellular level, different transporters are involved in the distribution and subsequently in the metabolism of BCAAs, with specificities differentiating epithelial and non-epithelial cells. At the subcellular level, there are also differences in transport throughout the different compartments. We will focus here on the heterodimeric transporters associating LAT1 (SLC7A5) or LAT2 (SLC7A8) and the glycoprotein CD98/4F2hc (SLC3A2) involved among others in the uptake of BCAAs in muscle and brain. LAT1 and LAT2 are antiporters that function through an exchange with glutamine. In the skeletal muscle (Figure 1), BCAA transport occurs in a tertiary mode: sodium-dependent uptake of glutamine via the SNAT1/2 transporters provides the glutamine necessary for the function of LAT1 and 2. This is of particular importance when considering the initiating role of glutamine in the activation by leucine of the mechanistic Target Of Rapamycin (mTOR) Complex 1 (mTORC1), the main effector of postprandial protein synthesis [6,7]. Note that transport via LAT1 and SNAT2 is influenced by insulin via different mechanisms such as increased protein expression (LAT1, SNAT2) or membrane translocation to the plasma membrane of intracellular storage vesicles (SNAT2) from a subcellular localization similar to that of the glucose transporter GLUT4 [8].

#### 2.2.2. BCAA Catabolism

Once imported into the cells, BCAAs can be used for protein synthesis. However, leucine, valine, and isoleucine are also metabolized via transamination by BCAA aminotransferase/transaminase (BCAT) into their respective BCKAs: α-ketoisocaproate (KIC), α-ketoisovalerate (KIV), and α-ketomethylvalerate (KMV) (Figure 2). There are two BCATs isoenzymes, a cytosolic one (BCATc or BCAT1) and a mitochondrial one (BCATm or BCAT2). BCAT1 is mainly found in neuronal tissues, where it plays a major role in glutamate synthesis (with α -ketoglutarate as the amino group acceptor) and nitrogen homeostasis [9]. It is also expressed in some immune cells, such as macrophages. Of note, BCAT1 seems to be involved in some aggressive tumors and could have a prognostic interest in these situations [10]. BCAT2 is found in most other tissues, except the liver (explaining the escape from the splanchnic territory); BCAA transamination thus occurs mainly in the mitochondria.

While, in the liver, part of KIC can be converted by a cytosolic dioxygenase into β-hydroxy-β-methylbutyrate (HMB), a ketone body [11]; BCKAs undergo mostly oxidative decarboxylation by BCKA dehydrogenase (BCKD) into branched-chain acyl-CoA, mainly leading to ketogenic and/or gluconeogenic derivatives. The BCKD complex, which shares similarities with pyruvate dehydrogenase and α-ketoglutarate dehydrogenase complexes (with a common dihydrolipoyl dehydrogenase E3), is associated with the matrix side of the inner mitochondrial membrane [12]. It represents the key controlling step of BCAA catabolism. It is subject to inactivating phosphorylation by BCKD kinase (BDK) [13] and activating dephosphorylation by the mitochondrial matrix-targeted protein phosphatase 2C (PP2Cm is also known as protein phosphatase Mg^2+^/Mn^2+^-dependent 1K (PPM1K)) [14], the expression of these two enzymes being regulated (respectively decreased and increased) by protein availability [15]. Leucine stimulates BCKD activity, promoting its dephosphorylation [16]. In fact, the activation of BCKD is rapidly achieved in all tissues through allosteric inhibition of its kinase BDK and activation of its phosphatase PP2Cm by KIC. In this respect, high dietary BCAA intake enables activation of BCKD for the rapid disposal of BCAAs and their keto-acids [15,17]. Thus, isolated excess leucine could lead to valine and isoleucine deprivation. This could explain why animal experiments have demonstrated that a very high (five-fold) enrichment in plasma leucine leads to a decrease in the level of the other BCAAs [18].

Following their metabolization by BCKD, branched-chain acyl-CoA are mainly degraded by their cognate enzymes into aceto-acetate and acetyl-CoA (leucine), propionyl-CoA, and acetyl-CoA (isoleucine) or propionyl-CoA (valine). Note that the penultimate step of leucine degradation is the degradation of 3-hydroxy-3-methylglutaryl-CoA by its lyase, the last enzyme of ketone body formation in the liver. In addition, propionyl-CoA is further metabolized into methylmalonyl-CoA and then into succinyl-CoA, feeding into the tricarboxylic acid (TCA) cycle. Last, some of the intermediates of this metabolism can serve as precursors of branched-chain and odd-chain fatty acids [19] or can escape the mitochondria as acyl-carnitines (C5 and C3 acylcarnitines), as hydroxyisobutyrate (HIB, valine catabolism), or as beta-amino-isobutyric acid (BAIBA, a product of the valine derivative methylmalonic semialdehyde) [20].

BCAA degradative metabolic processes are also closely related with energy metabolism. Under stress conditions, secretion of catabolic factors such as counter-regulatory hormones and inflammatory cytokines leads to peripheral protein catabolism. BCAAs released in this process are transaminated, their amino group being used for alanine and glutamine synthesis by the muscle and the BCKAs being oxidized for energy synthesis [21]. In highly catabolic states, such as the critically ill intensive care patient, BCAA oxidation in skeletal muscle can provide up to 20% of the energy used by muscles [22]. On the other hand, the transamination process confers to BCAAs an important role as nitrogen donors in some tissues/organs, as mentioned above concerning BCAT1. In the brain, BCAAs may be used for the synthesis of glutamate as an excitatory neurotransmitter [23].

In summary, BCAAs display a particular profile among amino acids in terms of distribution and metabolism. From a finalistic point of view, this is probably due, apart from their character as essential amino acids, to their importance in postprandial protein synthesis as substrates, but also, for leucine, as regulators of this synthesis (see below). In addition, they are very good substrates of energy metabolism, either at the muscular level or, when their muscular use reaches its limits, at the hepatic level, particularly for ketoneogenesis and gluconeogenesis in the post-absorptive period. Finally, the organism must take care to limit their plasma concentration, taking into account their potential central effects [24], namely their influence on the synthesis of neuromediators (via glutamate synthesis, or by competition with the cerebral transport of aromatic amino acids via LAT1/2 [8]) or because of their neurotoxicity, with the ultimate situation of maple syrup urine disease (MSUD, a BCKD deficiency) [25].

## 3. Metabolic Properties of BCAAs/BCKAs

Since the first demonstration in 1972 by Odessey and Goldberg [26], numerous studies have confirmed that BCAAs, and more precisely leucine, can stimulate protein synthesis and decrease protein catabolism. Leucine stimulates postprandial protein synthesis, mainly via activation of mTORC1 and the up-regulation of the cap-dependent mRNA translation initiation (Figure 1) [27,28]. The core of mTORC1 is formed by mTOR and two other proteins, Regulatory associated protein of mTOR (Raptor), and mammalian Lethal with SEC13 protein 8 (mLST8). To act, mTORC1 must undergo a Raptor-mediated translocation to the membrane of lysosomes. Leucine activates this translocation by binding to the Sestrin 2 protein, enabling Gator 2-Gator 1 interaction, activation of lysosome membrane-associated proteins (Rag), and binding of Raptor. Note that this activation requires leucine accumulation in, and efflux from, lysosomes, a mechanism conditioned by nutrient availability and inhibited in starvation conditions [29]. As a consequence, mTOR and the whole mTORC1 complex interact with the G protein Ras homolog enriched in brain (Rheb), enabling the activation of mTORC1 signaling. Interestingly, it has been shown that under conditions of increased nutrient availability, there is increased localization of endosomes/lysosomes near the cell membrane and anabolic signaling [30,31], increased lysosomal expression of LAT1 for leucine entry into lysosomes [32] and leucine efflux via the SLC38A9 transporter for mTOR activation [33].

Once active, mTORC1 phosphorylates 70-kDa ribosomal protein S6 kinase (p70S6K1) and eIF4E-binding protein 1 (4E-BP1), resulting in the stimulation of the initiation and elongation steps of protein synthesis. In turn, p70S6K1 also phosphorylates the inhibiting serine residue of insulin receptor substrate (IRS1), blocking insulin signaling and thus providing a negative feedback loop on insulin action [34,35]. mTORC1 also acts on the autophagy pathway, decreasing protein degradation via inhibitory phosphorylation of Unc-51-like kinase 1 (ULK1) complex and its downstream effector in autophagosome formation, the phosphoinositol 3-kinase vacuolar protein sorting 34 (VPS34) complex. Experiments with transaminase inhibitors have shown that the leucine effect on protein catabolism also partially depends on its degradation into KIC; however, the exact mechanism of this KIC effect is unknown. It should be noted that HMB, derived from KIC, is also an activator of protein synthesis and an inhibitor of muscle proteolysis, which has led to its proposed use in the treatment of muscle wasting in situations such as cancer cachexia, sarcopenia of aging, or critically ill patients [36]. However, its endogenous synthesis is far too low to contribute to the effects of leucine.

While experimental studies in rodents have clearly shown that leucine can both stimulate protein synthesis via the mTORC1 pathway and inhibits protein breakdown, its effectiveness in humans was discussed [37]. Indeed, most studies have demonstrated a decrease in protein degradation but limited effect on protein synthesis [21,37], and only a few studies have reported an effect on protein anabolism: for example, Fujita et al. [38] demonstrated that the administration of a single bolus of a leucine-enriched (35%) essential amino acids (0.35 g/kg fat free mass) and carbohydrate (0.5 g/kg fat free mass) mixture in young healthy volunteers led one hour later to a significant increase in muscle protein synthesis (4-fold) and the activation of mTOR signaling pathways (mTOR phosphorylation, p70S6K1, 4E-BP1). Katsanos et al. [39] showed that, in the elderly, leucine-enrichment of an essential amino acid supplement was associated with improved muscle protein synthesis. Thus, in a few studies, leucine seems to also activate protein synthesis in humans.

The anabolic effects of BCAAs can also be explained in part through their action on insulin secretion. Indeed, a physiological supply of leucine and isoleucine (1 to 2 g/kg) induces insulin secretion, an effect which decreases with ageing [40]. This insulin secretory effect has been attributed to the activation of glutamate dehydrogenase by leucine in pancreatic ß-cells, thus providing substrates such as α-ketoglutarate for the citric acid cycle (anapleurosis), which fuels the production of ATP. An increase in ATP cellular levels impacts the function of several channels at the level of the cell membrane, which ultimately leads to Ca^2+^ entry and exocytosis of insulin-loaded vesicles. Another non-exclusive mechanism could be a negative crosstalk between leucine metabolism and the activity of AMP kinase because AMP kinase has been shown to antagonize amino acid signaling and to decrease insulin production [41].

Another important aspect of these metabolic effects is the interaction between amino acids and glucose metabolism. Indeed, high physiologic concentrations of amino acids (and not specifically of leucine), while they stimulate protein synthesis, are associated with the inhibition of glucose metabolism [42]. Indeed, in healthy volunteers, euglycemic hyperaminoacidemic hyperinsulinemic clamp may be associated with reduced whole-body glucose disposal [43].

In summary, BCAAs, namely leucine, play an important regulatory role on protein homeostasis via (i) mTORC1 signaling and the activation of postprandial protein synthesis, (ii) an only partially elucidated inhibitory effect on proteolysis, and (iii) potentiation of insulin secretion. These muscular anabolic properties of leucine have led to the development of supplementation protocols in many conditions of muscle loss, but with varying outcomes [2,28]. These broad applications of leucine raise several issues, including the risk of developing insulin resistance in people supplemented with BCAAs, given the amino acid-induced reduction of whole-body glucose disposal. This question is all the more important because, as we will now see, and even if different factors are likely to increase the plasma concentrations of BCAAs, many studies show an association between the increase in these concentrations and insulin resistance, or even the risk of insulin resistance.

## 4. Blood BCAAs as Biomarkers of Metabolic Status

### 4.1. Physiological and Pathological Variations in Blood BCAAs

Different factors can influence plasma amino acid concentrations and, as with other essential amino acids, BCAA concentrations reflect a balance between inflow (protein intake and proteolysis) and outflow (oxidation and protein synthesis). In fact, at similar exogenous supplies of BCAAs, there is an increase in plasma BCAAs in all conditions associated with a decrease in oxidation and/or protein synthesis or an increase in peripheral protein catabolism, and a decrease in BCAAs in the opposite situations. A number of early studies have shown variations in plasma concentrations of BCAAs in different circumstances. For instance, Felig et al. [44], in 1969, observed an increase not only in BCAAs but also in aromatic amino acids, sulfur amino acids, threonine, and alanine in subjects with obesity compared to healthy controls. Studies from the same period also showed an increase in the plasma concentration of BCAAs during starvation, both in healthy and obese subjects [45], or in fasting type 1 diabetes patients at a distance from the administration of insulin [46]. An increase in plasma leucine and isoleucine, albeit to a more limited extent, has been demonstrated in healthy volunteers after 40 min of high-intensity exercise [47]. Plasma BCAAs also increase after injury, in particular if the injury is severe [48]; however, it has been shown to be decreased in severe sepsis. Other clinical situations, such as liver or kidney disease and aging, may also be associated with either increased or decreased plasma BCAAs [22].

Even in an apparently healthy population, it is possible to identify different factors affecting BCAA levels. Indeed, the analysis of more than 19,000 participants of the Women’s Health Study with no history of diabetes, cardiovascular disease, or cancer, showed that the highest concentrations of BCAAs were associated with higher age, higher BMI, lower physical activity, and unbalanced diet [49]. In addition, women in the highest quartile of BCAAs had a higher inflammatory status and a poorer plasma lipid profile. An additional cross-sectional analysis [50] indicated that the BMI explained 11.6% of BCAA variability. In a study in 487 obese patients with (homeostatic model assessment insulin resistance index (HOMA IR) > 2.5) or without insulin resistance, Guizar-Heredia et al. [51] described an association between plasma BCAAs and age (higher leucine below 30 years old). A meta-analysis of three genome-wide association studies of plasma BCAAs identified five genomic regions, the strongest association being with a sequence upstream of the BCKD phosphatase PPM1K gene. As expected, these increases in plasma BCAAs were associated with decreased oxidation by BCKD [52]. Note that in the study of Guizar-Heredia et al. [51], in obese patients, the non-common blood BCAA-raising alleles BCAT2 rs11548193 and BCKDH rs45500792 were associated with higher BCAA levels, and the difference was maintained after adjustment for BMI and insulin resistance.

As mentioned above, a factor potentially responsible for variations in BCAA levels, highlighted by epidemiological studies, is diet, including protein consumption. Interestingly, some human studies show differences between animal and plant proteins. In the study of Guizar-Heredia et al. [51], BCAA levels were positively correlated with animal protein intake and negatively with that of vegetable protein. Hamaya et al. [50], in their cross-sectional analysis of the Women’s Health Study, also showed an association of plasma BCAAs with animal but not vegetable protein intake. Note that Rousseau et al. [53] showed higher animal protein intake in their overweight subjects with metabolic syndrome compared to subjects with normal weight or overweight without metabolic syndrome. BCAA content of vegetal proteins displays a large variability but is generally either similar or significantly lower than that of animal protein (Table 1) [54]. Thus, the intake of BCAAs is likely to be somewhat lower with a diet rich in plant proteins. An additional confounding factor is the association found in these epidemiological studies between increased BCAAs and poor dietary quality index, i.e., a diet favoring simple sugars and fats, particularly saturated fats, and thus promoting insulin resistance.

In summary, several physiological and, most importantly, pathophysiological conditions are associated with an increase in blood BCAA concentrations. While initially the role of increased blood BCAA concentrations was not clear, important studies were published in late 2000 pointing to a possible causative role of BCAAs in insulin resistance, in particular the metabolomic study of Newgard et al. [55].

### 4.2. Blood BCAAs as Markers of Insulin Resistance

Newgard et al. [55] carried out a study including 73 obese subjects compared to 67 control subjects; these authors showed a strong association between insulin resistance (HOMA-IR) and a score based on plasma concentrations of BCAAs, but also methionine, glutamate, glutamine, phenylalanine, tyrosine, and C3 and C5 acylcarnitines. Similar associations between BCAAs and impaired glucose tolerance or diabetes have also been demonstrated by others [56,57]. An association between BCAAs and insulin resistance has been found in obese children, as shown in the systematic review of Zhao et al. [58] of 10 studies and 2673 children. Interestingly, Menni et al. [59] suggested, based on a nontargeted metabolomics study, that it was probably not the BCAAs themselves that were associated with diabetes, but their metabolism as BCKAs were significantly higher in subjects with impaired fasting glucose (fasting glucose: 5.6 to 7 mmol/L) or type-2 diabetes (T2D). They further showed that methyl-ketovalerate, the BCKA metabolite of valine, was a strong predictor for impaired fasting glucose. These results point to the importance of the alterations in the metabolism of BCAAs in this association with insulin resistance rather than the concentrations of the BCAAs themselves.

There are, however, some discrepant observations in the literature. For example, Hamley et al. [60] looked at young (18–35 years), nonobese (body mass index (BMI) < 30 kg/m^2^) Australian adults and measured glucose and insulin response during an oral glucose tolerance test (OGTT) with [6.6-^2^H]glucose administration. Individuals were categorized according to their insulin response and the existence or not of impaired fasting glucose or impaired glucose tolerance. These authors did not observe any increase in plasma BCAA concentrations in groups with significantly altered insulin sensitivity. Beyond the fact that, for example, study conditions may be much less stringent in epidemiological studies than in clinical ones, a confounding factor may be how insulin resistance is defined (fasting glucose, HOMA index, IV glucose tolerance test, euglycemic clamp, etc.). 

Unfortunately, experimental studies (Table 2) have not clarified this issue. Indeed, She et al. [61] in genetically-induced obesity models (ob/ob mice and Zucker rats) only observed a significant increase in plasma BCAA concentrations in the postprandial state or in “random fed” animals, but there was a near normalization in fasted animals. In models of diet-induced obesity (DIO), Newgard et al. [55] did not observe changes in Wistar rats fed a HF diet (19% protein, 35% carbohydrates, 45% fat) for 13 weeks. In Sprague-Dawley rats, while David et al. [62] observed an increase after approximately 7 weeks on a 60% fructose diet (19.3% protein, 64% carbohydrate, 16.7% fat), we never observed changes in BCAAs in Sprague-Dawley rats fed on either a 45%-fructose diet (19.3% protein, 64% carbohydrate, 8.4% fat) or a Western diet (20% protein, 35% carbohydrates, 45% fat diet, 30% fructose in drinking water) [63,64]. Last, in several long-term feeding experiments of DIO in C56Bl6 mice, a high-fat–high-sucrose diet (HFHS, 17% protein, 43% carbohydrates, 40% fat) was not associated with change in plasma BCAAs [65,66].

A number of interventional and experimental studies have investigated the influence of BCAA or protein intake on plasma BCAAs. These data must be analyzed with caution in terms of extrapolation of animal data to humans given the differences in metabolism and the nature of the experimental models (diet-induced or genetically induced obesity), the type of diet administered (Table 2), or the amount of BCAAs provided (often resulting in a doubling of plasma concentrations, without any relevance to the clinical reality). In the above-mentioned Newgard’s study [55], Wistar rats were fed for 13 weeks a conventional or a HF diet supplemented (150% higher content) or not with BCAAs. BCAA supplementation led to a doubling of their plasma concentrations in HF-fed animals, but only an approximately 35% increase in the control animals. Solon-Biet et al. [67] fed C57Bl6 mice for at least 15 months with standard rodent chow (18% total protein, 64% carbohydrate, 18% fat) but varying BCAA content (0.2, 0.5, 1 or 2-fold control level) and indeed observed that plasma levels increased with intake but reached a plateau at control level supply. Concerning humans, besides the numerous studies that have focused on the anabolic properties of leucine, often within short term trials, Woo et al. [68] performed, in 12 obese patients with prediabetes (fasting blood glucose: 100 to 126 mg/dL or glycated hemoglobin: 5.7 to 6.4%), a randomized cross-over study of supplementation with 20 g of BCAAs or low BCAAs proteins for 4 weeks; they showed that BCAA supplementation did not affect plasma BCAAs. Hattersley et al. [69] evaluated the effect of an 18-week increase in the supply of BCAAs given as protein (15, 20–25, or 25–30% of energy intake) with a constant fat intake (30%) in overweight or obese patients. Here also, plasma BCAAs were not modified. 

Nevertheless, keeping in mind the different confounding factors mentioned, a large number of metabolomic studies confirm an association between increased plasma BCAAs and insulin resistance. Interestingly, several cohort studies with follow-ups of 5 to 7.5 years indicate that BCAA levels may be predictive of the deterioration of insulin sensitivity and the development of T2D. Out of the 6244 subjects of the PREVEND (Prevention of renal and vascular end-stage disease) cohort, 301 developed T2D during a 7.5-year (median) follow-up [70]. In this cohort, Flores-Guerrero et al. [70] showed that patients in the highest BCAA quartile had an increased risk for developing T2D and that the association was maintained after correction for sex, age, BMI, and HOMA-IR. Later on, it was shown in a large subset of the same cohort that both high BCAA levels and fatty liver index (a score based on plasma triglycerides and gamma-glutamyltransferase activity, BMI and waist circumference) were predictive of incident T2D with respective hazard ratios of 1.19 (95% CI 1.03–1.37, *p* = 0.01) and 3.46, (95% CI 2.45–4.87, *p* < 0.001) [71]. Lotta et al. [52] meta-analyzed the data from five prospective studies (with a total of 1.992 incident T2D cases for a population of 6311 subjects) and also showed an increased risk of incident T2D linked to BCAA levels. This association was also true for BCKA, but not for metabolites downstream of BCKD in a subset of the Norfolk cohort of European Prospective Investigation into Cancer and Nutrition (EPIC-Norfolk). It should be noted that, here too, various factors can affect this relationship. Indeed, in the above-mentioned study of Guizar-Heredia et al. [51], there was an effect of age and insulin resistance on plasma BCAAs, resulting in similar BCAA concentrations between young patients without insulin resistance and those over 30 years with insulin resistance.

In the association between BCAAs and insulin resistance, we are then faced with two non-mutually exclusive possibilities:The decrease in insulin sensitivity affects the metabolism of BCAAs and induces an early increase in their plasma concentrations. This phenomenon is well known for other aspect of insulin resistance, for example in the cardiovascular field where alterations of the vascular function can exist many years before the appearance of clinical manifestations [72];The BCAAs or some of their derivatives could at certain doses or in certain situations of energy imbalance promote insulin resistance (e.g., through the increased serine phosphorylation of IRS1).

## 5. BCAA Metabolism and Insulin Resistance

### 5.1. BCAA Metabolism in Insulin Resistance Situations

It should first be noted that studies on the effects of insulin resistance on BCAA metabolism can sometimes seem inconsistent. It is therefore important to consider the specificities of this metabolism in humans and in rodents, but also the conditions under which the studies were carried out, in the fasted or fed state. Furthermore, although it is well established that there can be significant discrepancies between gene or protein expression, enzymatic activities, and substrate fluxes in vivo, it seems that this is particularly true for the metabolism of AACR [66].

#### 5.1.1. BCAA Metabolism and Insulin Resistance in Humans

It has been known for a long time that the peripheral utilization of amino acids in the postprandial period is closely related to the action of insulin. Already in the 1970s, Wahren et al. [5] showed, in insulin-dependent diabetic patients, in the absence of insulin, both a higher plasma concentration of BCAAs in the basal state and a greater increase in these concentrations in response to a protein meal. The measurement of leg postprandial arteriovenous amino acid differences showed that the postprandial increase in BCAA uptake was much more limited than in control subjects. In a study in 26 subjects, either overweight with impaired glucose tolerance (fasting glucose ≥ 5.6 mmol/L, 2 h glucose ≥ 7.8 mmol/L, or HOMA-IR > 2.0) or normal weight with normal glucose tolerance, mRNA-sequencing pathway analysis on subcutaneous white adipose tissue (WAT) and skeletal muscle biopsies indicated decreased post-prandial BCAA metabolism in both tissues, improved by a 12-week supervised exercise intervention [73].

In contrast, after an overnight fast, measurement of leucine turnover with 1-[^13^C]leucine or U[^13^C_6_]leucine showed a 9 to 50% increase in leucine flux and a 20 to 50% increase in leucine oxidation in subjects of increasing insulin resistance (HOMA-IR: 2.57 to 4.89) and normal to morbidly increased weight (BMI: 24.4 to 38.5) [74,75,76]. This suggests that, over the course of the day, the increase in leucine oxidation during fasting fails to compensate for the defect in postprandial utilization, which could lead to increased fasting plasma levels. Note that, in the Glynn et al. study [74], neither plasma BCAAs nor leucine flux were affected after a 6-month training program including aerobic and whole-body resistance training, despite a significant improvement in insulin sensitivity (mean 54% increase in glucose infusion rate during a hyperinsulinemic-euglycemic clamp). Similarly, plasma BCAAs and related metabolites were not affected by a 12-week supervised intensive exercise intervention [73]. On the other hand, when leucine oxidation was measured during a 2-step hyperinsulinemic-euglycemic clamp in obese subjects, while it increased as a function of insulin level in all subjects, it was lower in DT2 patients compared to non-DT2 ones [77]. In the liver, the decrease in hepatic BCAA oxidation is such that it is exploited in a breath test using ^13^C-KIC to evaluate the defect in hepatocyte mitochondrial function [78]. Indeed, Grenier-Larouche et al. [79] showed that the association between BCKA and nonalcoholic fatty liver disease (NAFLD), the hepatic manifestation of insulin resistance and metabolic symdrome, is the result of the defect in hepatic BCKA metabolism. In WAT, studies showed either preserved or decreased expression of enzymes involved in BCAA catabolism in subcutaneous or visceral WAT in obese subjects compared to normal weight individuals or in insulin-resistant obese compared to non-insulin-resistant patients [52,74]. Bariatric surgery was associated with increased BCKD expression. It can be noted, however, that measurements of BCAAs and BCKA fluxes from arteriovenous differences in subcutaneous adipose tissues in humans do not show any difference with obesity and/or insulin resistance [78]; but, this is not surprising given the small amplitude of the variations and the inter-individual variability.

#### 5.1.2. BCAA Metabolism in Experimental Models of Insulin Resistance

The defect in postprandial BCAA utilization is confirmed experimentally in models of insulin resistance with a more marked increase in post-prandial plasma BCAAs compared to controls [61]. Note that, in one study [52], the expression of the enzymes of BCAA catabolism in the fed state was not affected in the skeletal muscle but decreased in subcutaneous WAT; oxidation, only measured in WAT, was also decreased.

In the fasted state, experimental studies do not demonstrate any changes in protein expression of BCATm, BCKD E1α subunit, BDK, and BCKD E1 phosphorylation ratio [61,80], or BCKD activity [65] at the muscle level in models of genetic obesity (ob/ob mice, zucker rats) [61,80] or diet-induced (HF or HFHS) obesity [65,66]. In the liver, experimental data are discordant with, depending on the studies and models, normal or decreased expression of BCKD E1 subunit, normal or increased expression of BDK, and normal or increased BCKD E1 phosphorylation ratio [61,65,78]. However, both She et al. [61] and White et al. [81] observed a reduction in valine oxidation (using α-keto-[1-^14^C] isovalerate) in liver sample from ob/ob mice or zucker fatty rats, respectively. In WAT, studies are rather concordant. They globally observe a reduction in BCAA metabolism, whether in terms of enzyme expression or oxidation activity in subcutaneous or visceral adipose tissue in experimental models of genetic or diet-induced obesity [61,65,80,81]. Interestingly, treatment with thiazolidinediones in db/db mice was associated with an increase in protein expression of BCKD, suggesting a normalization of this metabolism.

Taken together, based on the data of whole-body leucine turnover and of organ BCAA metabolism (enzyme activities or expression), it seems that, in fasting insulin-resistant individuals, BCAA metabolism is characterized by (i) increased muscle oxidation and (ii) decreased oxidation in adipose tissue and perhaps in the liver. This is further illustrated by the study by Neinast et al. [82] extrapolating the contribution of different organs to BCAA metabolism from the distribution of ^13^C after administration of ^13^C-labeled BCAAs in mice. These authors show in models of insulin resistance in fasting animals, a redirection of the oxidation of BCAAs to the benefit of the muscles at the expense of WAT, and to a lesser extent of the liver.

#### 5.1.3. BCAA Metabolism in Brown/Beige Adipose Tissue

An interesting point highlighted by the above-mentioned work of Neinast et al. [82] as well as by that of Yoneshiro et al. [83] is the contribution of the brown/beige adipose tissue (BAT) in the oxidation of BCAAs. Long considered to be limited to hibernating mammals, various studies over the last two decades have highlighted the presence of a significant number of thermogenic adipocytes (brown or beige adipocytes) in humans and their potential contribution to energy metabolism. Indeed, Ouellet et al. [84] showed in healthy subjects exposed to cold, in conditions of limited shivering, an increase of up to 80% of resting energy expenditure, largely attributable to the brown/beige adipose tissue. If, until recently, it had been shown that this activation of BAT was associated with a significant oxidation of glucose and fatty acids, the work of Yoneshiro et al. [83] highlights an increased use also of BCAAs by the BAT in these conditions. In particular, these authors demonstrated an inverse relationship between BAT activity (assessed by ^18^F-Fluorodeoxyglucose positron emission tomography imaging) and plasma leucine and valine concentrations in healthy individuals. They further showed in mice the importance of BAT in the oxidation of BCAAs for thermogenesis and the maintenance of plasma concentrations of BCAAs. Finally, in a model of BCKD invalidation specifically in BAT, they showed that the defect in BCAA oxidation of the BAT was associated with weight gain and fat mass gain, increase in hepatic triglyceride content, and insulin resistance. Conversely, DIO in mice was associated with a decrease in BCAA catabolism in BAT [85]. Note that, in DIO mice, pharmacological stimulation of BCAA oxidation by 3,6-dichlorobenzo(b)thiophene-2-carboxylic acid (BT2) or by oral administration of some specific probiotics (*Bacteroides dorei* and *Bacteroides vulgatus*) improved BCAA catabolism by the BAT and mice metabolic status [85].

### 5.2. Do BCAAs Promote Insulin Resistance?

As already mentioned, leucine, through its interaction with Sestrin 2, promotes the activation of mTORC1, leading in particular to the activating phosphorylation of p70SK1 and inhibiting one of 4E-BP1, thus participating in the postprandial induction of protein synthesis. In essence, this system involves other regulatory elements or systems, for example, the availability of different substrates (e.g., via the level of amino acylation of tRNAs and General control nonderepressible 2 (GCN2)) or the level of restoration of protein pool. In terms of insulin signaling, through mTORC1 and p70S6K1, leucine induces a feedback inhibition via serine phosphorylation of IRS1. However, whether this feedback control causes insulin resistance is controversial. Alternatively, it has been proposed that it is not the BCAAs themselves but their metabolites that impair insulin sensitivity.

#### 5.2.1. BCAA-Induced Insulin Resistance?

While some experimental works seemed to support an effector role of BCAAs in insulin resistance, the results are quite inconsistent. In the above-mentioned Newgard et al. study [55], Wistar rats were fed for 13 weeks a conventional (C) or a HF diet supplemented (150% higher content) or not with BCAAs; BCAA supplementation lead to a doubling of their plasma concentrations. HF/BCAA-fed rats had lower food intake and lower weight gain than HF fed rats but had similarly impaired glucose tolerance, while C/BCAA-fed animals exhibited weight gain and glucose tolerance similar to those of C-fed rats. The authors acknowledged the responsibility of the BCAA/HF-diet combination.

These results contrast with, for example, those of Zhang et al. [86] in mice receiving, for 15 weeks, a doubled daily supply of leucine with either a control or a HF diet. They showed that, in HF-fed animals, a leucine-induced increase in energy expenditure was responsible for a reduction in weight gain and adiposity and was associated with improved glucose tolerance and lower insulin resistance. Indeed, leucine prevented the increase in blood glucose observed from the 4th week on the HF diet and improved insulin sensitivity throughout the study. In a more recent work [65], mice received an equivalent supplement of the 3 BCAAs (but half the amount of leucine) with an HF (12 weeks) or an HFHS (32 weeks) diet. Under these conditions, the BCAA supply did not affect glucose tolerance. A potential confounding factor could be the role of BCAAs in the regulation of food intake. In the Solon-Biet et al. study cited above [67], in which mice were fed a standard diet with variable BCAA intake, a double intake of BCAAs was associated with hyperphagia; this hyperphagia was related to reduced central serotonin synthesis and corrected by tryptophan supplementation. In addition, some of the differences between these studies, apart from the animal species, can be accounted for by different nutritional regimen (Table 2) with for example, according to the studies, quantitatively (40, 45, or 60% of the energy intake) and qualitatively (lard, milk fat, soybean oil) different lipid supplies or the supply of BCAAs individually or in mixture, in addition to the protein supply or in substitution of a part of the protein supply [55,67,86].

**Table 2 nutrients-15-00068-t002:** Nutritional conditions in some of the cited experimental studies.

Reference	Diet Type	Lipids	Protides	Carbohydrates
[61]	SC	18% 100% soybean oil	24% 100% soybean protein	58.0% 100% wheat/corn
[55]	HF	45% 12.3% soybean oil 87.7% lard	19% 53% casein 47% free amino acids	35% 21% corn starch 29% maltodextrin 50% sucrose
	HF/BCAA	43% 12.3% soybean oil 87.7% lard	23% 41% casein 58% free amino acids including BCAA supplement	34% 21% corn starch 29% maltodextrin 50% sucrose
	SC	10% of kcal 55.5% soybean oil 44.5% lard	19% of kcal 53% casein 47% free amino acids	71% of kcal 45% corn starch 5% maltodextrin 50% sucrose
	SC/BCAA	10% 55.5% soybean oil 44.5% lard	23% 41% casein 58% free amino acids including BCAA supplement	67% 45% corn starch 5% maltodextrin 50% sucrose
[62]	SC or FD	16.7% 100% soybean oil	19.3% 100% casein	64.0% 100% corn starch or fructose
[63]	SC	8.4% 100% soybean oil	19.3% 100% casein	72.4% 66% corn starch 33% maltodextrin
	FD	8.4% 100% soybean oil	19.3% 100% casein	72.4% 8% corn starch 92% fructose
[64]	HFHS	45% 20% soybean oil 80% lard	20% 100% casein	35% 73% rice starch 27% sucrose
		+30% fructose in drinking water		
[65]	SC	16.8% 100% lard	26.8% Mixed sources	56.4% 93% starch 7% sucrose
	HFHS	40% 95% milk fat 5% corn oil	17% 100% casein	43% 10% corn starch 20% maltodextrin 70% sucrose
	HF	54.8% 94.5% hydrogenated vegetable oil 5.5% corn oil	21.2% 100% casein	24% 69.6% corn starch 30.4% sucrose
	+ BCAAs in drinking water when added			
[66]	SC	13.5% 100% lard	28.5% mixed sources	58% 84.8% starch 9.8% sucrose 5.5% lactose
	HFHS	45% 87.7% lard 12.3% soybean Oil,	20% 100% casein	35% 20.8% corn starch 28.6% maltodextrin 50.6% sucrose
[67]	SC	18% 100% soybean oil	18% 100% casein	64% 63.5% wheatstarch 20.7% maltodextrin 15.7% sucrose
	BCAA adapted by reducing casein and adding free amino acids with or without BCAAs			
[86]	SC	13.1% 100% soybean oil	24.5% mixed sources	62.4% 91.2% starch 8.8% sucrose
	HF	60% 10% soybean oil 90% lard	20% 100% casein	20% 64% maltodextrin 36% sucrose
	+ BCAAs in drinking water when added			

Expressed as % energy and as % of each macronutrient. FD: fructose diet; HF: high fat diet; HFHS: high fat high sucrose diet; SC: standard chow.

In humans, previous studies showed a decrease in glucose utilization during amino acid infusion. Hyperinsulinemic euglycemic clamp tests have since been used to try to clarify these observations. For example, Boden and Tappy [87] investigated glucose metabolism in healthy volunteers receiving an infusion of a parenteral amino acid formula, leading to a 5- to 6-fold increase in their plasma concentrations during hyperinsulinemic euglycemic clamps associated with somatostatin and glucagon (0.25 and 3 g/kg/min). No changes in glucose metabolism were observed. On the contrary, under similar conditions, Krebs et al. [43] showed a 25% reduction in whole body glucose disposal. In fact, at least in short-term experiments, the responsibility of the global protein intake rather than the specifically of BCAAs seems likely. Indeed, both Smith et al. [88] and Harris et al. [89] have compared in sedentary obese women during hyperinsulinemic euglycemic clamps the effect of a protein supply or of an isomolar amount of leucine. Ingestion of proteins, but not leucine, significantly decreased insulin-stimulated glucose disposal. However, the previous studies were based on short-term evaluation. In longer-term studies, we also find this difference between the effect of the global protein intake and the effect of BCAAs. The type of proteins administered is involved; indeed, Pal et al. [90] compared the effects of a 12-week supplementation with 27 g of casein, whey (rich in BCAAs), or glucose (control) in overweight patients and observed an improvement of the insulin sensitivity with the whey supplementation. More specifically, essential amino acids do not seem to affect insulin sensitivity by themselves [91]. Finally, Woo et al. [68] compared in a crossover study the effects of a 4-week supplementation of 20 g of BCAAs or rice protein (low in BCAAs) in 12 obese prediabetic subjects and showed a trend towards improved glucose utilization with BCAAs.

Contrary to a possible BCAA-induced insulin resistance, several studies have demonstrated or suggested, particularly in the short term, an anabolic effect of BCAAs and more particularly of leucine. This notion, which has made them successful among sport enthusiasts in search of muscle gain, is based on the demonstration of an acute but transient activation of muscle protein synthesis. In the clinical setting, past studies have evaluated the possible benefit of BCAA supplementation in injured or cancer patients with, however, limited success, probably owing to methodological problems such as very inadequate parenteral nutrition mixtures not providing all the essential amino acids [2,22]. Since then, various studies have focused on the geriatric population with the aim of preventing or treating age-associated loss in muscle mass and function, i.e., sarcopenia. If short-term studies are globally positive, studies on prolonged periods are rather disappointing [28]. Interestingly, in healthy older adults (69.1 ± 1.1 yr) in a situation of prolonged bed rest (7 days of bed rest and 5 days of rehabilitation), leucine supplementation tended to preserve insulin sensitivity and muscle mitochondrial metabolism [92]. It should be noted that, because the various strategies taken individually have only limited effectiveness, a multimodal approach of sarcopenia is currently favored by combining leucine with another nutrient, such as vitamin D, or with resistance training exercise. For example, it has been shown in healthy elderly subjects that the intake of leucine and vitamin D for 6 weeks was associated with a significant increase in muscle protein synthesis and a gain in muscle mass [93]. The studies that evaluated insulin sensitivity in this context showed an improvement; however, this could have been expected given the improvement in body composition and in muscle mass, an important determinant of glucose disposal; therefore, it is difficult to relate it to a direct effect of leucine [94,95].

If BCAAs are responsible for insulin resistance, we can imagine that, conversely, a reduction in plasma BCAA concentrations could be associated with an improvement in insulin sensitivity. It should be noted that the studies on this subject were carried out in the context of the evaluation of the effects of caloric restriction and, more specifically, protein restriction, with very significantly reduced protein intake, lower than 50% of standard levels. It is clearly established, mainly on the basis of experimental work, that protein restriction improves metabolic health and longevity. Part of the effects seems related, at the endocrine level, to the decrease in the secretion of IGF-1 and the increase in the secretion of FGF-21 [96], a fasting and metabolic stress-induced peptide hormone that regulates energy substrates utilization [97]. At the cellular level, this involves the response of the mTOR and GCN2 pathways to the decrease in the global or specific availability of certain amino acids [96]. In Solon-Biet et al.’s study [67], median lifespan was similar between control and BCAA-restricted (at 20 or 50% level of standard supply) mice, but the 20% group had lower hepatic steatosis and lower basal insulin level with normal basal glucose. Moreover, Yu et al. [98] showed that a 2/3 reduction in protein supply or specifically in BCAA supply was associated with an improvement in glucose tolerance and a slowing down of weight gain after 3 weeks. Surprisingly, only an equivalent reduction specifically in isoleucine supply reproduced similar effects, while leucine restriction had no effect. The authors showed that the effects of isoleucine restriction were independent of the activation of the mTORC1 and GCN2 pathways in the liver and were associated with increased FGF-21 secretion, activation of ketogenesis, and increased energy expenditure. This was related to a specific, isoleucine deprivation-associated increase in nuclear FOXA2 (Forkhead box protein A2), a transcription factor that activates lipid metabolism and ketogenesis and whose nuclear localization is blocked by insulin [99]. The effect of BCAA restriction has been reproduced in T2D patients receiving a one-week supply of proteins and amino acids (1 g/kg body weight) restricted (−60%) or not in BCAAs [100]; BCAA restriction was associated with an increase in the plasma concentration of FGF-21 and in the “oral glucose sensitivity index”, calculated during a “mixed meal tolerance test”.

In summary, the results of the restriction studies show a specific response to BCAA deficiency that illustrates the importance of these amino acids in both protein synthesis and ketogenesis and the necessary adaptation of the metabolism for a better management of energy sources. Observations in excess situations are very contrasted beyond the simple postprandial regulation of substrate excess; the multiple confounding factors (animal species, nature of the associated diet, activity level of the animals, etc.) suggest that the direct role of BCAAs in insulin resistance is quite limited.

#### 5.2.2. BCAA Metabolites and Insulin Sensitivity

From metabolomic or epidemiological studies, some authors have hypothesized that it was probably not BCAAs themselves but their metabolites that could contribute to insulin resistance. Indeed, She et al. [101] observed that global invalidation of BCATm, which induces a blockade of BCAA degradation and an increase in their plasma concentrations, was associated with a reduced weight and fat mass gain, improved insulin sensitivity, and resistance to the obesogenic effect of a HF diet. In the same way, telmisartan, an angiotensin II receptor blocker with unique BCATm inhibitory properties, favors adipose tissue browning and improves body weight, glucose tolerance, and insulin sensitivity in HF-fed mice [102]. Of course, these data must be interpreted with caution as BCAAs/BCKAs are significant contributors to energy metabolism in situations where glucose/glycogen availability is limited and significant regulators of protein homeostasis. The blockade of their degradation must therefore be compensated by metabolic adaptation, as already mentioned in situations of severe protein or BCAA restriction. Nonetheless, the metabolism of BCAAs could be necessary to allow their negative effects on energy metabolism. Indeed, some metabolomic and epidemiological studies showed associations between plasma concentrations of certain metabolites and insulin resistance or the risk of developing insulin resistance. The presence of some of these metabolites, mainly BCKA, C5 and C3 acylcarnitines, and HIB, exacerbated in experimental models during concomitant administration of BCAAs and HF diet, has been interpreted as an indication of incomplete oxidation of BCAAs, and it has been suggested that these metabolites are potentially responsible for insulin resistance. Note that, as in the case of fatty acid oxidation and the appearance of even acyl-carnitines [103], increased BCAA derivative levels, particularly odd acyl-carnitines, may not be indicative of incomplete oxidation but may simply indicate an increase in metabolic flux. This is well illustrated by the study of energy substrate utilization during aerobic exercise as a function of glycogen availability. Under conditions of reduced glycogen availability, the muscle increases the oxidation of lipid substrates (with no change in exogenous carbohydrate utilization) but also of BCAAs from protein catabolism [104,105,106]. Metabolomic analysis showed an increase in fatty acid metabolites such as acyl-carnitine proportionately to the increase in lipid oxidation; similarly, the increase in BCAA metabolites was proportional to the increase in leucine oxidation [105]. In insulin-resistant subjects, as we have seen, BCAAs are metabolized more slowly in the postprandial period, but their oxidation increases in the post-absorptive period. Therefore, it does not seem surprising that the plasma concentrations of the reactive intermediates increase during these same periods.

The negative effect of BCKAs on insulin function is suggested, for example, by the improvement of glucose tolerance and insulin sensitivity in ob/ob or DIO mice induced by increasing BCKD activity following BDK inhibition by BT2 [107]. In vitro work in C2C12 or L6 muscle cells gives some indication of a potential direct role of BCKAs in insulin resistance. In these cell models, BCKAs decrease Akt phosphorylation and thus insulin signaling and decrease glucose utilization [66,108]. More precisely, Biswas et al. [66] showed inhibition of insulin induced activating tyrosine phosphorylation of IRS1 (at very high KIC concentration) but upregulated protein translation signaling and synthesis. This is similar to the previously mentioned effect of leucine on the Akt-mTORC1 pathway, but this conclusion remains controversial. In addition, it has been shown that valsartan, an angiotensin II receptor blocker that displays BDK inhibitory properties [109], elicited only limited effects on insulin sensitivity in human. Indeed, the Navigator study on 9306 patients with impaired glucose tolerance and cardiovascular disease showed very limited effect of valsartan on glucose tolerance [110]. In fact, another mechanism may explain some of the experimental effects of BDK inhibitors. Indeed, the effect of BDK inhibition by BT2 is already observed after one week of treatment in Zucker rats and is associated with a shift in respiratory exchange ratio, suggesting decreased fatty acid storage and increased oxidation [111]. White et al. [111] showed that BDK and PPM1K also control ATP-citrate lyase phosphorylation, and thus de novo lipogenesis and fat storage. Interestingly, in a mouse model of nonalcoholic steatohepatitis (NASH), pharmacological inhibition of ATP-citrate lyase reduced hepatic steatosis, fibrosis, inflammation, blood glucose, and triglycerides [112], suggesting that the beneficial effect of BT2 may also be ascribed to this ATP-citrate lyase inhibition rather than to BCKD activation.

Data in support of a role for acylcarnitines are quite limited [103]. In C2C12 myotubes, Aguer et al. [113] evaluated the effects of C4 acylcarnitines on insulin signaling and showed a significant reduction in Akt phosphorylation, which contrasts with the presumed effects of BCAAs that lead to hyperactivation of the Akt/mTORC1 pathway. On the contrary, and although this evidence is indirect, various studies show rather absent or inverse relationships. Experimentally, in HFHS- or HF-fed mice, BCAA supplementation is associated with increased C3 and C5 acyl-carnitine, with no impact on insulin resistance. From a clinical point of view, in the follow-up of patients with NAFLD, C5 acylcarnitine concentrations decreased with the progression of liver damage [114]. In the metabolic syndrome, Remchak et al. [115] characterized the chronotype of their patients and showed, in the early chronotype group (“early birds”) compared to the late chronotype group (“night owls”), a better insulin sensitivity and an increase in plasma concentrations of most acylcarnitines.

The last candidate is HIB [103]. Note that HIB-dehydrogenase deficiency, a rare metabolic disease with inability to metabolize HIB, does not seem associated with severe insulin-resistance [116]. Moreover, efforts to improve insulin sensitivity by increasing the flux through the BCKD are a priori accompanied by an increase in the availability of HIB by simple flux effect. The hypothesis of a detrimental effect of HIB is based on the work of Jiang et al. [117], showing that conditioned-medium from C2C12 myoblasts overexpressing peroxisome proliferator-activated receptor-gamma coactivator 1α (PGC-1α) contained increased levels of HIB and that this HIB increased trans-cellular transport of fatty acids by HUVECs (human umbilical vein endothelial cells) for subsequent oxidation. However, in vivo, although mice supplemented with high doses of HIB displayed delayed glucose disposal, they presented a normal basal blood glucose. These data are all the more puzzling because PGC-1α activation has been associated with improved metabolic status. In this respect, Roberts et al. [118], in myocyte overexpressing PGC-1α, showed an increase in BAIBA release and that this BAIBA induced adipocyte browning. In addition, Hatazawa et al. [119] did not observe any increase in muscle HIB in PGC-1α transgenic mice but an increase in BAIBA content.

In summary, there is really very limited data to support a role for acyl-carnitines or HIB in the development of insulin resistance. With respect to BCKAs, the control of ATP-citrate lyase by the same kinase and phosphatase that control BCKD requires further study, particularly in terms of the interaction between these two metabolic pathways upon exposure to a diet high in both lipids and BCAAs.

## 6. Conclusions

A chronic increase in plasma concentrations of BCAAs is a predictive marker for the risk of developing insulin resistance and subsequently T2D. The fact that this can be observed long before the deterioration of glucose tolerance testifies to the existence of early asymptomatic insulin resistance and the role of insulin in the metabolism of BCAAs. The fact that there is a competition between amino acids and glucose to the detriment of glucose in situations of excess availability of amino acids is probably a testimony to the limited capacity of the organism to store amino acids, a limit reached as soon as the protein pool is restored. The priority then becomes the elimination of these excess amino acids and the conservation of glucose.

Insulin resistance will only worsen in populations that are increasingly exposed to a sedentary lifestyle, poor dietary balance, and environmental insults. Excessive energy supply (even if limited, but over the long term) and sedentary lifestyle will reinforce this defect in BCAA utilization due to the progressive reduction of their oxidation in adipose tissue and compromised muscle mass and the lack of oxidation by brown adipose tissue. The reduced postprandial utilization of BCAAs, due to insulin resistance and the reduction of the anabolic response to the meal, is partly compensated by an increased oxidation in the post-absorptive period, especially as these amino acids are very good ketogenic substrates. The consequence is an increase in plasma concentrations of intermediates of BCAA degradation in the post-absorptive period. The evidence for a possible role of these intermediates on insulin resistance is at this stage quite limited.

## Figures and Tables

**Figure 1 nutrients-15-00068-f001:**
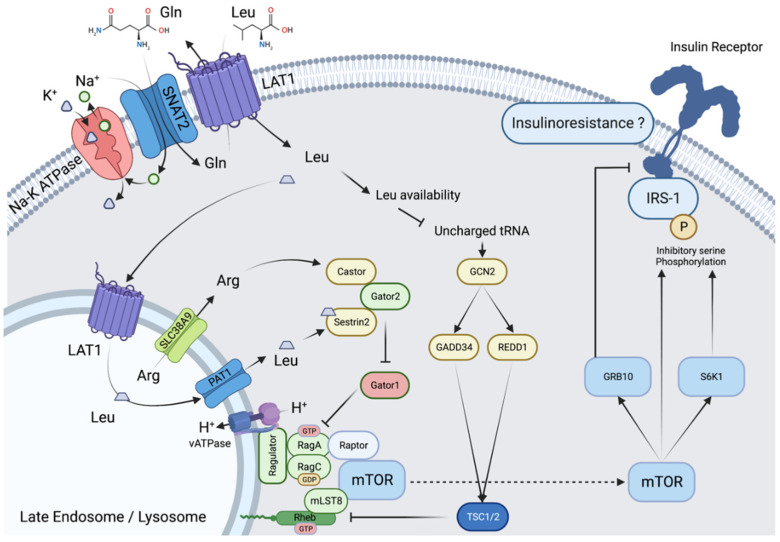
Activation of mTORC1 by leucine in muscle cells. Activation of mTORC1 in muscle cells involves (i) counter-transport in exchange with glutamine and transport into endosomes, localized near the cell membrane, via LAT1, (ii) increased leucyl-tRNA charging inhibiting the GCN2 pathway, (iii) leucine efflux from the endosome via PAT1 involving inward proton transport via vATPase, (iv) interaction of leucine with the protein sestrin2, lifting the inhibitory effect of the GATOR1/2 complex on RAG-GTPases and allowing (v) localization of mTORC1 to the endosome and its activating interaction with Rheb.

**Figure 2 nutrients-15-00068-f002:**
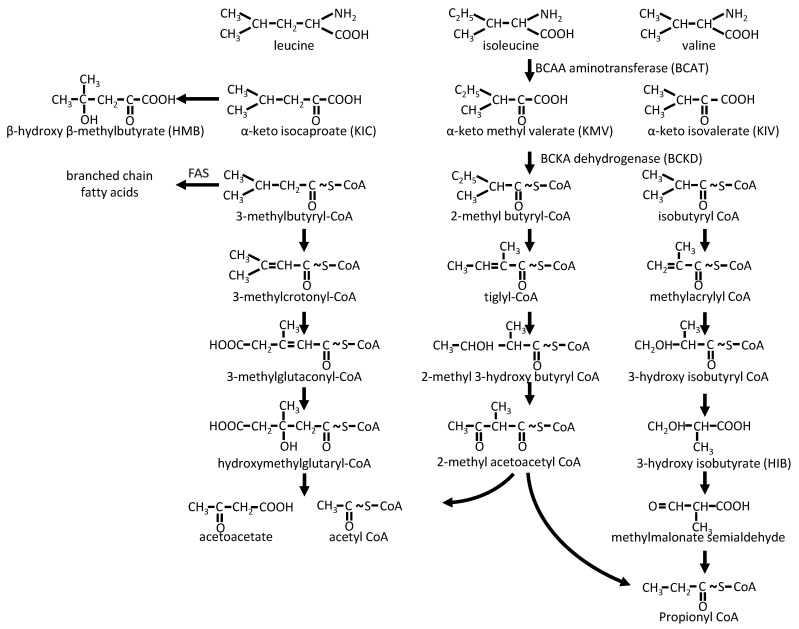
BCAA catabolic pathways. FAS: Fatty acid synthase.

**Table 1 nutrients-15-00068-t001:** BCAA content of different foods.

		Leu			Val			Ile			BCAAs		
		Energy ^a^	Protein ^b^	Weight ^c^	Energy ^a^	Protein ^b^	Weight ^c^	Energy ^a^	Protein ^b^	Weight ^c^	Energy ^a^	Protein ^b^	Weight ^c^
ANIMAL	Chicken	1.45	7.98	2.50	0.96	5.28	1.66	1.02	5.62	1.76	3.42	18.88	5.92
	Beef	1.27	8.26	2.12	0.79	5.15	1.32	0.73	4.73	1.21	2.79	18.14	4.65
	Pork	1.08	8.17	2.31	0.73	5.52	1.56	0.63	4.77	1.35	2.43	18.46	5.21
	Salmon	1.19	8.43	2.16	0.75	5.34	1.37	0.67	4.78	1.23	2.61	18.54	4.76
	Egg	0.66	8.55	1.02	0.47	6.10	0.73	0.42	5.46	0.65	1.54	20.10	2.39
	Whole milk	0.46	8.89	0.28	0.31	6.13	0.19	0.25	4.85	0.15	1.02	19.86	0.62
LEGUMES	Peas	0.53	7.50	0.62	0.35	4.94	0.41	0.30	4.32	0.36	1.18	16.76	1.39
	Lentils	0.62	7.89	0.72	0.43	5.40	0.50	0.37	4.70	0.43	1.42	17.99	1.65
	Soybeans	0.69	7.70	1.19	0.42	4.72	0.73	0.41	4.58	0.71	1.52	17.00	2.64
	Black beans	0.57	8.43	0.75	0.37	5.52	0.49	0.32	4.65	0.42	1.26	18.60	1.66
GRAINS	Whole wheat	0.29	6.78	0.87	0.19	4.41	0.57	0.15	3.63	0.47	0.63	14.82	1.90
	Brown rice	0.19	8.28	0.21	0.14	5.85	0.15	0.10	4.22	0.11	0.42	18.35	0.47
	Oats	0.26	7.76	0.22	0.18	5.45	0.16	0.13	3.98	0.11	0.57	17.20	0.49
	Quinoa	0.27	7.13	0.32	0.19	5.04	0.23	0.16	4.28	0.19	0.61	16.45	0.75
NUTS	Almonds	0.24	6.53	1.42	0.14	3.79	0.82	0.12	3.33	0.72	0.51	13.64	2.96
	Peanuts	0.29	6.43	1.65	0.19	4.16	1.07	0.16	3.49	0.89	0.64	14.07	3.61
	Pecans	0.09	6.73	0.63	0.06	4.62	0.43	0.05	3.78	0.36	0.21	15.13	1.42
	Walnuts	0.18	7.42	1.16	0.11	4.78	0.75	0.10	3.97	0.62	0.39	16.16	2.53
VEGETABLES	Potatoes	0.12	6.01	0.10	0.11	5.60	0.09	0.08	4.08	0.07	0.30	15.69	0.26
	Carrots	0.13	5.95	0.06	0.09	4.03	0.04	0.10	4.49	0.04	0.33	14.47	0.13
	Butternut	0.15	6.88	0.06	0.12	5.26	0.05	0.10	4.72	0.04	0.37	16.87	0.15
	Sweet potatoes	0.14	6.19	0.12	0.13	5.77	0.12	0.08	3.67	0.07	0.35	15.63	0.32
	Green beans	0.41	7.62	0.14	0.33	6.11	0.11	0.24	4.53	0.09	0.98	18.25	0.34
FRUITS	Apple	0.03	6.47	0.02	0.03	5.97	0.02	0.02	2.98	0.01	0.08	15.42	0.04
	Banana	0.10	7.97	0.09	0.07	5.51	0.06	0.04	3.28	0.04	0.21	16.76	0.18
	Mango	0.10	7.40	0.06	0.08	6.21	0.05	0.06	4.29	0.03	0.24	17.90	0.14
Mushrooms		0.83	6.05	0.18	1.60	11.70	0.35	0.52	3.83	0.12	2.95	21.58	0.65

Adapted from [54]. ^a^ kcal/100 kcal; ^b^ g/100 g prot; ^c^ g/100 g product.

## Data Availability

Not applicable.

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
