# Peer review of "Branched-Chain Amino Acids and Insulin Resistance, from Protein Supply to Diet-Induced Obesity"

_nutrients, 2022, doi:10.3390/nu15010068_

Round 1
Reviewer 1 Report
Thank you for the opportunity to review your work.
General Comment
The presented article presents an interesting topic, but the methodology, and conclusions should be better explained and clarified to be publishable.
It is necessary to explain the methodology used to carry out the literature review.
In point 2, there is a lack of coherence in the text because it jumps from one paragraph to another changing the subject without a proper transition.
Line 235 The summary should specify that there is little evidence of this in humans
Point 4. Improvements should be made to the wording of the first sentences.
Line 270. The name of the study should be capitalized as is done later when it is mentioned again.
Line 411. The reference should be indicated after the authors' names.
Conclusions
The conclusions should be improved to reflect the main findings of the review; it makes no sense to mention the pollutant issue if it has not been mentioned and analyzed in depth previously in the review.
Author Response
We thank reviewer 1 for his constructive comments. We have modified the manuscript except for the third point.
It is necessary to explain the methodology used to carry out the literature review.
This review is the result of a long-term bibliographic survey on branched-chain amino acids and insulin resistance as it is now indicated in the manuscript. The keywords used were: branched-chain amino acid OR leucine OR valine OR isoleucine AND insulin resistance. This is now mentioned in the introduction
In point 2, there is a lack of coherence in the text because it jumps from one paragraph to another changing the subject without a proper transition.
Sorry for that; this section has been reorganized
Line 235 The summary should specify that there is little evidence of this in humans
With all due respect, we disagree. This has been demonstrated in humans. Without citing all the studies (e.g., ref 38 in the manuscript), it has been reviewed by us (ref 29 in the manuscript) or by others (see for example Borack and Volpi J Nutr. 2016 Dec; 146(12): 2625S–2629S)
Point 4. Improvements should be made to the wording of the first sentences.
The paragraph has been modified
Line 270. The name of the study should be capitalized as is done later when it is mentioned again.
This has been corrected
Line 411. The reference should be indicated after the authors' names.
This has been changed
The conclusions should be improved to reflect the main findings of the review; it makes no sense to mention the pollutant issue if it has not been mentioned and analyzed in depth previously in the review.
This has been modified
Reviewer 2 Report
The manuscript by Jean-Pascal De Bandt., et al., titled as “Branched-chain amino acids and insulin resistance, from protein supply to diet-induced obesity” is a reviewed manuscript. Authors have extensively studied and reviewed the existing research about BCAA’s and its metabolites effect on insulin resistance, type2 diabetes, obesity relation to protein intake and plasma conc.
Here are my comments:
1. Authors have extensively reviewed about leucine only. Failed to provide literature for valine and isoleucine.
2. BCAA’s have role in liver metabolism, fatty liver. Authors could have mentions central metabolic organ role in BCAA’s metabolism.
3. Table 1: What is E, P, W? and add please point/dot (.) not comma (,) between values.
4. As per the provided data in table 1. 100gm of chicken can only give 5.92g of BCAA’s. But the same diet cane gives higher amount of fats and cholesterol. Is BCAA’s have synergic or compound effect (insulin secretion/plasma levels) with increased sugars and fats intake?
5. Authors have provided huge amount of text; it would be better to have illustrative figures about BCAA’s and its metabolites effects on systemic organs and final outcomes.
6. Section 5.1.3 at line 413, authors have discussed about only on BAT, why not WAT and other adipose tissues. Recent literature like BCAA–BCKA axis regulates WAT browning through acetylation of PRDM16 can give more information.
Author Response
We thank reviewer 2 for his constructive comments. We have modified the manuscript except for the fifth point.
- Authors have extensively reviewed about leucine only. Failed to provide literature for valine and isoleucine.
As now indicated in the text, this review is the result of a long-term bibliographic survey on branched-chain amino acids and insulin resistance. The keywords used were: branched-chain amino acid OR leucine OR valine OR isoleucine AND insulin resistance
As it happens, the literature is very unbalanced since the works of the 80's showing an exclusive anabolic effect of leucine and therefore a consequent number of mechanistic or clinical studies is devoted to this anabolic effect of leucine alone. Apart from these studies, few of the other studies have focused exclusively on leucine. For example, as shown in the table, many experimental studies vary the supply of all BCAAs or test the three BCAAs administered separately. Another example: among the BCAA derivatives suspected of inducing insulin resistance, C3 and C5 acyl carnitines can be derived from all 3 BCAAs while experiments to modulate transaminase or dehydrogenase activity involve all 3 BCAAs. Finally, hydroxybutyrate is a derivative of valine.
- BCAA’s have role in liver metabolism, fatty liver. Authors could have mentions central metabolic organ role in BCAA’s metabolism.
We thank the reviewer for this remark, which highlights an inaccuracy in our manuscript. As described in section 2, the liver has only very limited BCAT activity and thus does not metabolize BCAA. On the other hand, as described in section 5.1, hepatic BCKA metabolism is most likely decreased. This decrease in hepatic oxidation is indeed exploited in a breath test using 13C-ketoisocaproate to assess hepatocyte mitochondrial function (see for example Afolabi et al. J Breath Res. 2018;12:046002). The association between BCKA and fatty liver disease is thus the result of the defect in hepatic BCKA metabolism (see for example Grenier-Larouche et al. JCI Insight. 2022;7:e159204) The manuscript has been modified (see section 5.1.1) to emphasize this point.
- Table 1: What is E, P, W? and add please point/dot (.) not comma (,) between values.
Sorry for the lack of clarity. As mentioned in the table legend: kcal/g, g/g protein and g/g product. This has been modified
- As per the provided data in table 1. 100gm of chicken can only give 5.92g of BCAA’s. But the same diet cane gives higher amount of fats and cholesterol. Is BCAA’s have synergic or compound effect (insulin secretion/plasma levels) with increased sugars and fats intake ?
Epidemiological studies do not seem to have search for these interactions and, as described in section 4.2, experimental studies gave conflicting results. This issue is further complicated by the fact that the increase in plasma BCAA also correlates with poor diet quality score. A sentence has been added in the manuscript (see section 4.1).
- Authors have provided huge amount of text; it would be better to have illustrative figures about BCAA’s and its metabolites effects on systemic organs and final outcomes.
We agree, but the diversity of the literature and the protocols used makes it very difficult to summarize the systemic effects in a figure (unlike the cellular and molecular effects of leucine on the mTORC pathway in Figure 1). We actually regret not being able to produce this figure at this stage of a knowledge that still needs to be improved.
- Section 5.1.3 at line 413, authors have discussed about only on BAT, why not WAT and other adipose tissues. Recent literature like BCAA–BCKA axis regulates WAT browning through acetylation of PRDM16 can give more information.
In fact, the "classic" data on adipose tissue, whether visceral or subcutaneous, are discussed, within the limits of available data, in sections 5.1.1 and 5.1.2. Brown or beige adipocytes are a new player in the BCAA metabolism equation and therefore deserved a separate place. We have modified this section to better highlight this aspect.
The effect of telmisartan, to which the reviewer refers (BCAA-BCKA axis regulates WAT browning...) is mentioned at the beginning of section 5.2.2 (ref 102). This underlines the complexity of the issue since we observe beneficial effects of the inhibition of BCAA catabolism (e.g. by telmisartan) but also of the acceleration under the action of BT2 of their catabolism which will provide more of these acetyl-CoA susceptible to acylate PRDM16.
Reviewer 3 Report
Comments to the Author
In my opinion, article requires general improvement. After corrections it may be reconsidered.
1. Complete the abbreviations used in the text of the article in the abstract.
2. Correct the quotation - it should be at the end of the sentence.
3. Review the literature as required by the journal.
Author Response
We thank reviewer 3 for his comments and have modified the manuscript accordingly.
- Complete the abbreviations used in the text of the article in the abstract.
This has been added
- Correct the quotation - it should be at the end of the sentence.
This has been corrected wherever necessary
- Review the literature as required by the journal.
The strategy of the review is now indicated (see section 1)
Round 2
Reviewer 1 Report
The review carried out by the authors is gratefully acknowledged, it should simply be added the time during which the search was carried out and whether they applied inclusion or exclusion criteria to the articles used in the review.
Author Response
We thank the reviewer for his comment. Time during which the search was carried out, article exclusion criteria and additional searches are now indicated.